# Mutation Analysis of Autosomal-Dominant Polycystic Kidney Disease Patients

**DOI:** 10.3390/genes14020443

**Published:** 2023-02-09

**Authors:** Yasuo Suzuki, Kan Katayama, Ryosuke Saiki, Yosuke Hirabayashi, Tomohiro Murata, Eiji Ishikawa, Masaaki Ito, Kaoru Dohi

**Affiliations:** 1Department of Cardiology and Nephrology, Mie University Graduate School of Medicine, Tsu 514-8507, Japan; 2Department of Kidney center, Suzuka Kaisei Hospital, Suzuka 513-8505, Japan; 3Department of Nephrology, Saiseikai Matsusaka General Hospital, Matsusaka 515-0003, Japan

**Keywords:** autosomal-dominant polycystic kidney disease (ADPKD), exome, *PKD1*, *PKD2*, mutation

## Abstract

Autosomal-dominant polycystic kidney disease (ADPKD) is characterized by bilateral kidney cysts that ultimately lead to end-stage kidney disease. While the major causative genes of ADPKD are *PKD1* and *PKD2*, other genes are also thought to be involved. Fifty ADPKD patients were analyzed by exome sequencing or multiplex ligation-dependent probe amplification (MLPA), followed by long polymerase chain reaction and Sanger sequencing. Variants in *PKD1* or *PKD2* or *GANAB* were detected in 35 patients (70%). Exome sequencing identified 24, 7, and 1 variants in *PKD1*, *PKD2*, and *GANAB*, respectively, in 30 patients. MLPA analyses identified large deletions in *PKD1* in three patients and *PKD2* in two patients. We searched 90 cyst-associated genes in 15 patients who were negative by exome sequencing and MLPA analyses, and identified 17 rare variants. Four of them were considered “likely pathogenic” or “pathogenic” variants according to the American College of Medical Genetics and Genomics guidelines. Of the 11 patients without a family history, four, two, and four variants were found in *PKD1*, *PKD2*, and other genes, respectively, while no causative gene was identified in one patient. While the pathogenicity of each variant in these genes should be carefully assessed, a comprehensive genetic analysis may be useful in cases of atypical ADPKD.

## 1. Introduction

Autosomal-dominant polycystic kidney disease (ADPKD) is characterized by progressive enlargement of bilateral kidney cysts, which ultimately leads to end-stage kidney disease [1]. Extrarenal manifestations include liver cysts, valvular heart disease (VHD), and brain aneurysm [2]. The prevalences of liver cysts, VHD, and brain aneurysm in ADPKD patients are reported to be approximately 83%, 25%, and 10%, respectively [3,4,5].

While the main causative genes of ADPKD are *PKD1* and *PKD2*, *GANAB* is also known as *PKD3* [6]. The genetic analysis of *PKD1* is complicated in comparison to *PKD2* or *GANAB* because there are six *PKD1* pseudogenes, from *PKD1P1* to *PKD1P6,* as well as *PKD1* on chromosome 16 [7]. Long polymerase chain reaction (PCR) was used in previous genetic studies to differentiate true *PKD1* from the six *PKD1* pseudogenes [8,9,10,11,12,13].

Exome sequencing is a powerful tool to identify causative genes for chronic kidney disease (CKD); 31% of the genetically diagnosed cases are due to *PKD1* and *PKD2* mutations [14]. Although the causative genes of ADPKD are mainly *PKD1* and *PKD2*, many genes are associated with cystic kidney diseases, including nephronophthisis or autosomal-dominant tubulointerstitial kidney disease (ADTKD). Therefore, exome sequencing is superior to Sanger sequencing for performing a comprehensive analysis. A multiplex ligation-dependent probe amplification (MLPA) analysis of *PKD1* or *PKD2* is also useful for detecting large deletions or insertions [15,16].

Therefore, we have analyzed 50 ADPKD cases by exome sequencing or multiplex ligation-dependent probe amplification (MLPA), followed by long PCR and Sanger sequencing.

## 2. Materials and Methods

### 2.1. Patients

Patients who were being treated in the outpatient ward between March 2020 and June 2022 were enrolled in the present study. The inclusion criteria were as follows: age >20 years and >5 kidney cysts in each kidney according to abdominal computed tomography (CT) or magnetic resonance imaging (MRI). The exclusion criteria were as follows: <20 years of age, ≤5 five kidney cysts in each kidney according to abdominal CT or MRI, or lack of informed consent. After obtaining written informed consent from 50 patients, blood samples were collected. Hypertension was defined as blood pressure >130/80 mmHg or taking an anti-hypertensive drug. Total kidney volume (TKV) was calculated with 3D construction or an ellipsoid equation using the length, width, and depth of the kidneys.

### 2.2. Exome Sequencing

Genomic DNA was extracted from blood samples of the 50 patients using a Blood Genomic DNA Extraction Mini Kit (FAVORGEN, Vienna, Austria). The DNA concentration was measured with a Qubit dsDNA HS Assay Kit (Thermo Fisher Scientific, Waltham, MA, USA). The Ion AmpliSeq Exome RDY Library (Thermo Fisher Scientific) was prepared according to the manufacturer’s instructions. The library concentration was examined with a QuantStudio 3D digital PCR system (Thermo Fisher Scientific). Templates were prepared with an Ion PI Hi-Q Chef Kit (Thermo Fisher Scientific) and sequenced with an Ion Proton System (Thermo Fisher Scientific). Variants in the 93 genes (*AHI1, ALG8, ANKS6*, *ARL13B*, *ARL6*, *ARMC9*, *ASS1*, *B9D1*, *B9D2*, *BBIP1*, *BBS1*, *BS2*, *BBS4*, *BBS5*, *BBS7*, *BBS9*, *BBS10*, *BBS12*, *C5orf42*, *C8orf37*, *CC2D2A*, *CCDC28B*, *CEP41*, *CEP83*, *CEP104 CEP120*, *CEP164*, *CEP290*, *CSPP1*, *DCDC2*, *DNAJB11*, *DZIP1L*, *GANAB*, *GLIS2*, *HNF1B*, *IFT27*, *IFT43*, *IFT74*, *IFT122*, *IFT140*, *IFT172*, *INPP5E*, *INVS*, *IQCB1*, *KIAA0556*, *KIAA0586*, *KIF14*, *KIF7*, *LRP5*, *MAPKBP1*, *MKKS*, *MKS1*, *MUC1*, *NEK8*, *NOTCH2*, *NPHP1*, *NPHP3*, *NPHP4*, *OFD1*, *PDE6D*, *PIBF1*, *PKD1*, *PKD2*, *PKHD1*, *PRKCSH*, *REN*, *RPGRIP1L*, *SDCCAG8*, *SEC61A1*, *SEC61B*, *SEC63*, *SLC41A1*, *SUFU*, *TCTN1*, *TCTN2*, *TCTN3*, *TMEM67*, *TMEM107*, *TMEM138*, *TMEM216*, *TMEM231*, *TMEM237*, *TRAF3IP1*, *TRIM32*, *TSC2*, *TTC21B*, *TTC8*, *UMOD*, *WDPCP*, *WDR19*, *WDR35*, *XPNPEP3*, and *ZNF423*) [17,18,19,20,21,22,23,24,25,26,27,28] were examined.

### 2.3. Multiplex Ligation-Dependent Probe Amplification (MLPA) Analyses

MLPA analyses were performed with a SALSA MLPA kit P351 PKD1 or SALSA MLPA kit P352 PKD1-PKD2 (MRC Holland, Amsterdam, The Netherlands) to detect deletions or duplications in the *PKD1* or *PKD2* gene, respectively. The results were analyzed with Coffalyser.Net (MRC Holland, Amsterdam, the Netherlands).

### 2.4. Sanger Sequencing

Long polymerase chain reaction (PCR) was performed with KOD FX Neo (Toyobo, Tokyo, Japan) according to the manufacturer’s instructions. The reaction volume was 25 μL, which included 12.5 μL of 2×PCR buffer for KOD FX Neo, 5 μL of 2 mM dNTP mix, 2 μL (30–50 ng) of template DNA, 0.75 μL of 10 μM forward primer, 0.75 μL of 10 μM reverse primer, 0.5 μL of KOD FX Neo, and 3.5 μL of PCR-grade water. The PCR conditions were as follows: pre-denaturation at 94 °C for 2 min, 5 cycles of denaturation at 98 °C for 10 s and extension at 74 °C for 30 s per kilobase (kb), 5 cycles of denaturation at 98 °C for 10 s and extension at 72 °C for 30 s per kb, 5 cycles of denaturation at 98 °C for 10 s and extension at 70 °C for 30 s per kb, 25 cycles of denaturation at 98 °C for 10 s and extension at 68 °C for 30 s per kb, and extension at 68 °C for 7 min. Long PCR primers and sequencing primers for *PKD1* are shown in Appendix A [29]. Normal PCR was performed with HotStarTaq DNA Polymerase (Qiagen, Hilden, Germany). The PCR conditions were as follows: 96 °C for 15 min, 35 cycles of denaturation at 96°C for 45 s, annealing at 57 °C for 45 s, and elongation at 72 °C for 1 min and 72 °C for 15 min. PCR primers for *PKD2* and *GANAB* are shown in Appendix A. Sanger sequencing was performed with an Applied Biosystems 3130 Genetic Analyzer (Applied Biosystems, Waltham, MA, USA). The sequence results were analyzed with the BioEdit software program and compared using the Ensembl database.

### 2.5. Pathogenicity Evaluation

The pathogenicity of the identified variants was evaluated according to the American College of Medical Genetics and Genomics (ACMG) guidelines [30]. Previous reports of each variant were examined in the ClinVar database [31] and Leiden Open Variation Database (LOVD) [32]. The minor allele frequency (MAF) of each variant was searched in the Genome Aggregation Database (gnomAD) and 3.5KJPNv2 database, and rare minor allele frequency (MAF) was defined as <1% [33]. The pathogenicity of variants was assessed in silico with software programs such as Polymorphism Phenotyping v2 (PolyPhen-2) and Sorting Intolerant From Tolerant (SIFT).

## 3. Results

### 3.1. Background Data

Fifty patients were analyzed (Table 1). The mean age of the 50 patients was 56 ± 13 years, and 18 patients (36%) were male. A family history of ADPKD was observed in 39 patients (78%), and the number of other affected members is shown in Table 1. Microhematuria was observed in nine patients (18%). The median protein/creatinine ratio was 0.13 (0.07–0.39) g/g·Cr, and the protein/creatinine ratio was >0.5 g/g·Cr in nine patients (18%). The mean estimated glomerular filtration rate (eGFR) was 48.7 ± 27.3 mL/min/1.73 m^2^. The median TKV was 1266 (877–1716) ml. Twenty-six patients (52%) received tolvaptan at an average dose of 57 ± 31 mg. The Mayo classifications of the patients were as follows: class IA (*n* = 6), class IB (*n* = 6), class IC (*n* = 11), class ID (*n* = 4), and class II (*n* = 18) [34]. Five patients were unclassified because four patients were diagnosed with stage 5 CKD and one had an eGFR of >100 mL/min/1.73 m^2^.

### 3.2. Mutation Analyses of Patients with Autosomal-Dominant Polycystic Kidney Disease

#### Exome Sequencing

Exome sequencing was performed in 50 patients, and variants in *PKD1*, *PKD2,* or *GANAB* were detected in 30 patients (60%) (Table 2). Among the 30 patients, there were 24 variants in *PKD1*, 7 variants in *PKD2*, and 1 variant in *GANAB*; patients 1 and 10 had two variants in *PKD1*. Nineteen of the thirty-two variants were considered to be “likely pathogenic” or “pathogenic” according to the ACMG guidelines (Appendix A). Twenty-two of the thirty-two variants were unreported in the ClinVar or LOVD databases and were considered to be novel. To confirm variants in *PKD1*, long PCR followed by sequencing was performed (except for *PKD1* c.12671C>A in patient 1, c.12386T>A in patient 5, and c.11441_11459dup in patient 32, which were confirmed by Sanger sequencing). The 24 variants in *PKD1* were classified as follows: missense (*n* = 10), insertion frameshift (*n* = 4), deletion (*n* = 3 (2 were frameshifts)), near splice-site (*n* = 3), and nonsense (*n* = 4). Sanger sequencing was performed to confirm variants in *PKD2* or *GANAB*. The seven variants in *PKD2* were classified as follows: splice-site (*n* = 3), deletion frameshift (*n* = 2), missense (*n* = 1), and nonsense (*n* = 1). The variant in *GANAB* was a missense variant.

### 3.3. MLPA Analyses

MLPA analyses of *PKD1* or *PKD2* were performed in 20 patients who did not have any variants in *PKD1*, *PKD2*, or *GANAB* in exome sequencing and in 10 patients who had unreported missense variants (patients 5, 11, 20, 26, 30, 45, and 47) or intronic variants (patients 15, 17, and 18). Five novel large deletions in patients 4, 8, 14, 29, and 44 were identified (Figure 1A). In patients 8 and 44, the deletion of exons 31–34 of *PKD1* was observed, and Sanger sequencing identified a 1694-bp deletion between intron 30 and intron 34 in *PKD1*, leading to the deletion of exons 31–34 of *PKD1* (Figure 1B). The deletion of exons 2–6 in *PKD1* was observed in patient 29, and Sanger sequencing identified an 8354-bp deletion and an insertion of ATC between intron 1 and exon 7 in *PKD1*, leading to the deletion of exons 2–7 of *PKD1* (Figure 1B). The deletion of all exons of *PKD2* was observed in patients 4 and 14, and Sanger sequencing identified a 723424-bp deletion between intron 3 in *NUDT9* and intron 1 in *ABCG2* in patient 14 (Figure 1B), while a breakpoint could not be confirmed in patient 4.

### 3.4. The Exome Analysis of Cyst-Associated Genes

Rare variants with MAF <1% in the 90 genes were examined in 15 patients who showed negative results in exome sequencing and MLPA analyses. The results were confirmed by Sanger sequencing and are summarized in Table 3. The PCR primers are shown in Appendix A. In all, 19 variants were identified in 13 patients. While 17 were heterozygous, *OFD1* c.1215A>C in patient 7 was hemizygous, and *PKHD1* c.9629C>G in patient 28 was homozygous. Four of the nineteen variants were considered to be “pathogenic” or “likely pathogenic” according to the ACMG guidelines (Appendix A). Seven of the nineteen variants were unreported in the ClinVar or LOVD databases and were considered to be novel. Rare pathogenic variants in other genes were examined in 15 patients (Appendix A). There were no rare pathogenic variants in other genes in patients 28, 38, or 39.

Then, rare pathogenic variants in other genes in 11 patients (pt 1, 5, 11, 15, 17, 18, 20, 26, 30, 45, 47) with variants of uncertain significance in *PKD1*, *PKD2,* or *GANAB* were examined in exome sequencing (Appendix A). There were no rare pathogenic variants in other genes in patient 26.

### 3.5. Comorbidity Evaluation

The prevalences of liver cysts, VHD, and brain aneurysm are summarized in Table 4. Liver cysts were observed in 43 patients (86%), among whom 7 had severe liver cysts. Six of the seven patients with severe liver cysts were female. The causative genes of severe liver cysts were five variants in *PKD1*, one variant in *GANAB*, and one variant in *PKHD1*. Hypertension was observed in 40 patients (80%). The median value of brain natriuretic peptide (BNP) was 23.8 [13.1–38.2] pg/mL, and 30 patients (60%) had a value of >18.4 pg/mL. The mean ejection fraction (EF) in an ultrasound cardiography examination was 68.9 ± 6.6%, and 48 patients (96%) had a value of >50%. Mild regurgitation was observed at the aortic valve (*n* = 7), mitral valve (*n* = 23), tricuspid valve (*n* = 28), and pulmonary valve (*n* = 20). Moderate regurgitation was observed in the aortic valve (*n* = 5) and tricuspid valve (*n* = 1). Severe tricuspid valve regurgitation was observed in one patient. The mean E/A and E/e’ ratios were 1.0 ± 0.4 and 9.1 ± 2.9, respectively. Septal e’ < 7 cm/s, septal E/e’ > 15, tricuspid regurgitant velocity (TRV) > 2.8 m/s, and left atrial volume index (LAVI) >34 mL/m^2^ were observed in 21, 3, 0, and 2 patients, respectively. Patients 19 and 35 fulfilled two of the four criteria for heart failure with preserved ejection fraction (HFpEF), with increased BNP levels [35]. The causative genes of VHD were as follows: *PKD1* (*n* = 19), *PKD2* (*n* = 8), *GANAB* (*n* = 1), and other genes (*n* = 9); the causative gene was unknown in two patients. Brain aneurysms were observed in eight patients (16%), all of whom were female. The causative genes of brain aneurysm were three variants in *PKD1*, two variants in *PKD2*, and three variants in other genes. Thirty-seven patients had a low (score 0–3) Predicting Renal Outcome in Polycystic Kidney Disease (PROPKD) score, while thirteen had an intermediate (score 4–6) PROPKD score (Table 4 and Appendix A) [36].

Eighteen patients had a type II Mayo classification (Table 1); among these patients eight had “likely pathogenic” or “pathogenic” variants, including four *PKD1*/*PKD2* deletions and four other variants such as *DNAJB11*, *NEK8*, *PKHD1*, or *WDR19*. Transverse and coronal CT images of the eight patients are shown in Figure 2. The kidney images of patients 8, 14, 29, and 44 were similar (Figure 2A). There were no apparent liver cysts in patients 35 and 46 (Figure 2B). The kidney cysts in patient 35 were mostly seen in the kidney medulla. The kidney cysts and TKV in patient 38 were small. The kidney cysts in patient 41 had calcification. Each of the kidney cysts in patient 46 was relatively small.

### 3.6. Change in eGFR and TKV without and with Tolvaptan

Because tolvaptan was administered to 26 patients (52%), the change in eGFR and TKV was examined in patients who were followed up for more than one year (Table 5). Sixteen patients were managed without tolvaptan, and twenty-four were managed with tolvaptan. The mean eGFR changes per year in patients managed without and with tolvaptan were −2.6 ± 1.6 and −3.1 ± 2.6 mL/min/1.73 m^2^, respectively. The mean initial TKV values in patients managed without and with tolvaptan were 1059 ± 615 and 1429 ± 631 mL, respectively. The mean TKV changes per year in patients managed without and with tolvaptan were 3.8 ± 7.0% and 5.0 ± 5.5%, respectively.

## 4. Discussion

We demonstrated 32 variants in *PKD1*, *PKD2*, or *GANAB* in 30 of 50 patients (60%) in the exome analyses; 19 of these variants (59%) were considered to be “likely pathogenic” or “pathogenic” according to the ACMG guidelines, and 22 variants (69%) were considered to be novel. We also identified five novel large deletions in 5 of 20 patients (25%) in MLPA analyses. Additional exome analyses of 90 cyst-associated genes were conducted for 15 patients who showed negative results in exome sequencing and MLPA analyses; these identified 19 variants, of which 4 (21%) were considered to be “pathogenic” or “likely pathogenic” according to the ACMG guidelines, and 7 variants (37%) were considered to be novel. Overall, “likely pathogenic” or “pathogenic” variants were identified in 28 patients; these were identified by *PKD1*/*PKD2* exome analyses in 19 patients; MLPA analyses in 5 patients; and exome sequencing of 90 genes in 4 patients. Other patients had variants of uncertain significance, which did not indicate a definite diagnosis until further proof was obtained.

Of the 50 patients in the present study, the causative genes were *PKD1* in 25 patients (50%), *PKD2* in 9 patients (18%), and *GANAB* in 1 patient (2%). Variants in other cyst-associated genes that were identified in 13 of 50 patients (26%) were variable, including *BBS12*, *CEP164*, *DNAJB11*, *GLIS2*, *IFT140*, *KIF7*, *NEK8*, *OFD1*, *SLC41A1*, *TMEM67*, *WDPCP*, *WDR19*, and *ZNF423* in 1 patient, *NPHP3* in 2 patients, and *PKHD1* in 4 patients. No causative genes were identified in 2 of the 50 patients (4%). While rare pathogenic variants in other genes were examined in 15 patients who showed negative results in exome sequencing or MLPA analyses, their functional roles in ADPKD were undetermined. Rare pathogenic variants in other genes were also examined in 11 patients with variants of uncertain significance in *PKD1*, *PKD2*, or *GANAB* in a similar way, and while the functional roles in ADPKD were undetermined for most of these variants, the roles of *CPLANE1* in patient 18 and *AHI1* in patient 47, both of which were cilia-associated genes, might be involved in the development of ADPKD to some degree.

Of the 39 patients with a family history of ADPKD, the causative genes were *PKD1* in 21 patients, *PKD2* in 7 patients, *GANAB* in 1 patient, other genes in 9 patients, and unknown in 1 patient. On the other hand, the causative genes of the 11 patients without an apparent family history of ADPKD were variable, including 4 variants in *PKD1* (missense (*n* = 2), nonsense (*n* = 1), large deletion mutation (*n* = 1)), 2 variants in *PKD2* (frameshift deletion (*n* = 1), splice-site deletion mutation (*n* = 1)), 1 missense variant in *OFD1*, 1 missense variant in *WDPCP*, 1 missense variant in *CEP164*, 1 nonsense variant in *PKHD1*, and 1 unknown.

Liver cysts were observed in 43 patients (86%), and 7 of the 43 patients had severe liver cysts. The causative genes of the severe cysts were *PKD1* (nonsense (*n* = 2), large deletion (*n* = 1), frameshift insertion (*n* = 1), in-frame deletion (*n* = 1)), *GANAB* (missense (*n* = 1)), and *PKHD1* (missense (*n* = 1)). The causative genes in seven patients (14%) without liver cysts were *PKD1* (missense (*n* = 2), in-frame deletion (*n* = 1)), *OFD1* (missense (*n* = 1)), *PKHD1* (nonsense (*n* = 1)), *WDPCP* (missense (*n* = 1)), and *WDR19* (splice-site (*n* = 1)); genes other than *PKD1* or *PKD2* or *GANAB* were observed in four of seven patients (59%).

The high prevalence of hypertension in the present study (80%) might reflect the activation of the renin-angiotensin system or the sympathetic nervous system in ADPKD patients [37]. Thirty patients (60%) had increased serum BNP levels, which was compatible with studies reporting that serum BNP levels were increased in patients with CKD [38,39]. Two patients (4%) fulfilled two of the four criteria for HFpEF with an EF of >50% and an elevated serum BNP level. The causative genes in the two patients were *ZNF423* (missense variant) in patient 19 and *GLIS2* (missense), *PKHD1* (missense), or *WDR19* (splice-site) in patient 35. VHD was observed in 39 patients (78%), and the causative genes were *PKD1* (missense (*n* = 6), in-frame deletion (*n* = 2), frameshift deletion (*n* = 1), large deletion (*n* = 3), frameshift insertion (*n* = 3), near splice-site (*n* = 3), nonsense (*n* = 2)), *PKD2* (missense (*n* = 1), frameshift deletion (*n* = 2), large deletion (*n* = 1), splice-site (*n* = 3), nonsense (*n* = 1)), *GANAB* (missense (*n* = 1)), other genes in 10 patients, and unknown in 2 patients. The causative genes in 11 patients (22%) without VHD were *PKD1* (missense (*n* = 3), frameshift insertion (*n* = 2), nonsense (*n* = 1)), *PKD2* (large deletion), *PKHD1* (missense, nonsense), *CEP164* (missense), *KIF7* (missense), and *TMEM67* (missense).

Brain aneurysm was observed in eight patients (16%), and the causative genes were *PKD1* (missense (*n* = 2), nonsense (*n* = 1)), *PKD2* (missense (*n* = 1), splice-site (*n* = 1)), *ZNF423* (missense), *GLIS2* (missense) or *PKHD1* (missense) or *WDR19* (splice-site), and *CEP164* (missense).

Tolvaptan is frequently used in the treatment of ADPKD due to its effectiveness [40]. Patients with a TKV of >750 mL and with a growth rate of >5% are suitable for tolvaptan treatment. The goal of tolvaptan treatment is to slow the progression of kidney cysts, which can slow the progression of CKD in this disease. The present study showed a decline in eGFR of −3.1 mL/min/1.73 m^2^ per year and a 5.0% TKV change per year, which might be acceptable according to previous research [40]. Careful follow-up of the patients receiving tolvaptan treatment should be continued.

The present study was associated with the following limitations. In the present study, no variants were identified in patient 25 or 39. Patient 25 had stage 5 CKD, and his kidney cysts may have been related to hemodialysis treatment. Patient 39 had a relatively small TKV for her age, and the causative gene may be unknown. While two variants in *PKD1* were found in patient 1 and patient 10, respectively, we could not examine whether these acted as in cis or in trans variants, as blood samples could not be obtained from parents or siblings. While *DNAJB11* is a known disease-causing gene of ADPKD [25], and kidney cysts can develop in patients who are heterozygous for a pathogenic variant in *PKHD1*, more evidence is required to determine whether heterozygous *NEK8* and *WDR19* pathogenic variants can mimic the ADPKD phenotype. Rare pathogenic variants in other genes were not validated in Sanger sequencing. We did not examine variants in the introns of 93 genes in the present study.

## 5. Conclusions

While the pathogenicity of each variant in cyst-associated genes should be carefully assessed, a comprehensive genetic analysis may be useful for atypical ADPKD cases.

## Figures and Tables

**Figure 1 genes-14-00443-f001:**
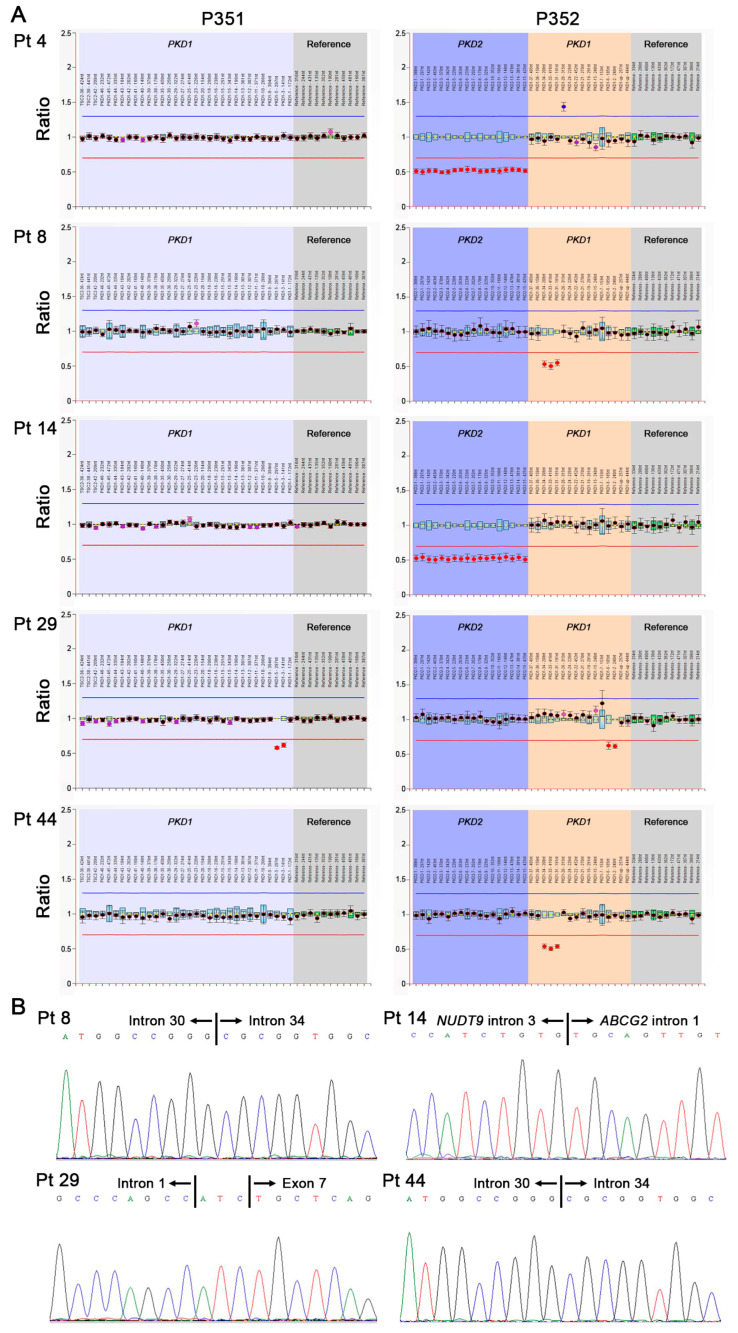
MLPA analyses. (**A**) Five large deletions in patients 4, 8, 14, 29, and 44 were identified. (**B**) Sanger sequencing identified deletions of exons 31–34 of *PKD1* in patients 8 and 44, a deletion of Iexons 2–7 of *PKD1* in patient 29, and a deletion between intron 3 in *NUDT9* and intron 1 in *ABCG2* in patient 14, while a breakpoint could not be confirmed in patient 4. Pt, patient.

**Figure 2 genes-14-00443-f002:**
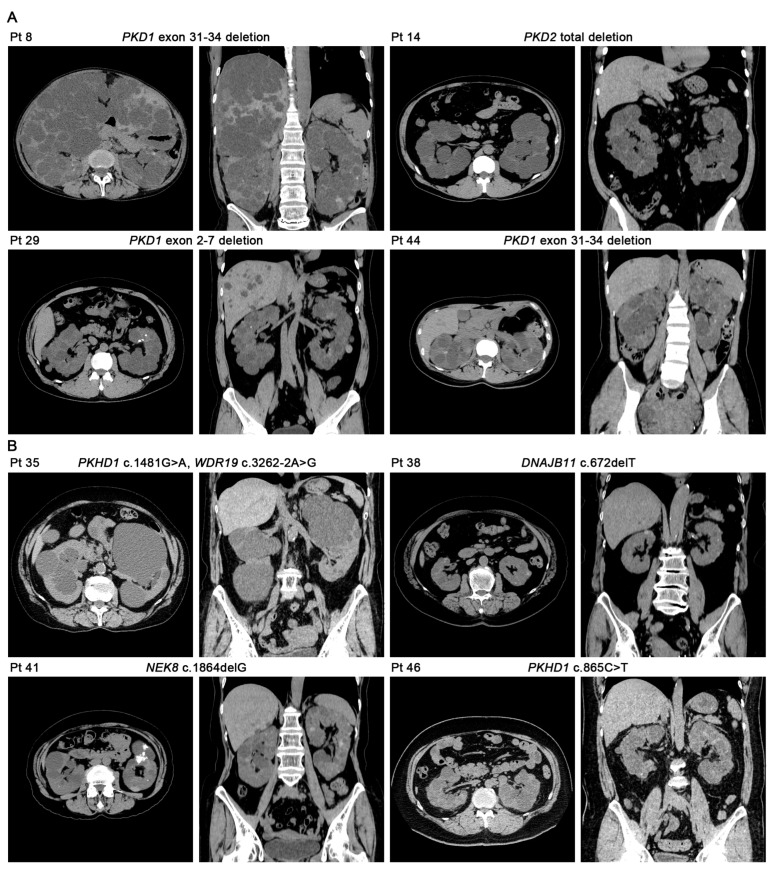
Transverse and coronal CT images of eight patients with “likely pathogenic” or “pathogenic” variants and Mayo type II classifications. (**A**) The kidney images in patients 8, 14, 29, and 44 were similar. (**B**) The kidney cysts in patient 35 were mostly seen in the kidney medulla, without apparent liver cysts. The kidney cysts and TKV in patient 38 were small. The kidney cysts in patient 41 had calcification. Each of the kidney cysts in patient 46 was relatively small without apparent liver cysts.

**Table 1 genes-14-00443-t001:** Clinical characteristics of the study subjects.

Pt	Age	Sex	Family History	Microhematuria	Proteinuria (g/g·Cr)	eGFR (mL/min/1.73m^2^)	TKV (mL)	Tolvaptan (mg)	Mayo (Class)
Pt 1	35	M	1	0	0.08	86.9	986	90	IC
Pt 2	65	F	2	0	0.09	28.2	973	60	IB
Pt 3	84	M	1	0	0	5.2	1384	0	NA
Pt 4	61	F	2	0	0	3	1524	0	NA
Pt 5	47	F	3	1+	0.13	58.6	1064	30	IC
Pt 6	70	M	0	2+	0.74	13.4	2549	0	NA
Pt 7	78	M	0	1+	0.09	59.5	859	0	II
Pt 8	51	F	2	0	0.67	20.2	874	45	II
Pt 9	54	F	2	0	0.35	41.9	2425	120	IC
Pt 10	53	M	2	0	0.25	45.1	2903	60	ID
Pt 11	60	F	2	2+	0.47	46.8	1346	15	IB
Pt 12	36	F	2	0	0.17	89.6	1363	0	ID
Pt 13	51	M	2	0	0.04	35.6	2332	0	II
Pt 14	52	M	2	0	0.84	31.8	1961	120	II
Pt 15	45	F	2	0	0.48	23.2	2079	75	IC
Pt 16	60	F	2	0	0.39	27.8	1752	45	IC
Pt 17	70	F	1	0	0.13	18.1	544	7.5	IA
Pt 18	43	F	1	0	0	68.7	1168	60	IC
Pt 19	79	F	1	(+/−)	0.54	56	503	0	II
Pt 20	59	M	0	0	0.08	55.9	1608	30	IC
Pt 21	64	F	4	0	0.07	25	1309	45	IB
Pt 22	51	F	1	0	0	40.2	1559	75	IC
Pt 23	47	M	0	0	1.63	18.1	2727	22.5	ID
Pt 24	48	F	0	1+	0.17	54.3	3841	45	ID
Pt 25	71	M	0	NA	NA	8	2908	0	NA
Pt 26	75	F	4	0	0.14	37	692	0	IA
Pt 27	55	M	0	0	0.12	45.3	1608	60	IC
Pt 28	58	F	1	0	0	70.4	586	0	II
Pt 29	38	M	0	0	0	74.5	1544	120	II
Pt 30	32	F	2	0	0.12	144.5	269	0	NA
Pt 31	41	F	2	0	0.25	98.9	396	0	IA
Pt 32	39	F	2	0	0.39	46.1	886	60	IC
Pt 33	58	F	1	(+/−)	0.22	66.4	518	0	IA
Pt 34	48	F	2	0	0.05	79.2	312	0	IA
Pt 35	69	F	1	0	0	78.7	2089	60	II
Pt 36	54	F	2	0	0	71.9	377	0	IA
Pt 37	77	M	0	2+	1.49	22.2	1408	0	II
Pt 38	70	F	2	(+/−)	0.05	47	402	0	II
Pt 39	56	F	3	0	0.09	40.7	678	0	II
Pt 40	58	M	1	0	0.58	22.9	1603	60	IC
Pt 41	67	F	1	0	1.4	62.4	908	0	II
Pt 42	54	M	1	0	0.98	21.1	1110	37.5	IB
Pt 43	51	M	1	0	0.11	64.9	2088	0	II
Pt 44	35	F	1	0	0.09	90.4	965	15	II
Pt 45	79	F	1	0	0.06	43.8	1094	0	IB
Pt 46	48	M	0	0	0.31	37.1	891	60	II
Pt 47	53	M	2	0	0.07	62.1	1222	0	IB
Pt 48	65	F	1	0	0.18	49.8	1441	0	II
Pt 49	69	F	3	0	0.18	44.5	1768	0	II
Pt 50	45	F	0	0	0.22	50.7	885	60	II

eGFR, estimated glomerular filtration rate; F, female; M, male; NA, not available; Pt, patient; TKV, total kidney volume.

**Table 2 genes-14-00443-t002:** Summary of exome sequencing.

Pt	Gene	Variants	Amino Acid Change	ACMG	ClinVar	LOVD	dbSNP	gnomAD	3.5KJPNv2	PolyPhen-2 (Score)	SIFT (Score)
Pt 1	*PKD1*	c.7302_7313delGCGGGGCGTGCT	p.Gly2436_Arg2439del	US	Unreported	Unreported	NA	NA	NA	NA	NA
Pt 1	*PKD1*	c.12671C>A	p.Thr4224Asn	US	Unreported	Unreported	rs200685883	0.0000217	0.00018	0.725	APF (0.03)
Pt 2	*PKD2*	c.965G>A	p.Arg322Gln	LP	LP	P2, LP1	rs145877597	0.000003982	NA	1	APF (0.00)
Pt 3	*PKD1*	c.1831C>T	p.Arg611Trp	LP	P1, LP2, US3	P1, LP1	rs1555458413	NA	NA	1	APF (0.00)
Pt 5	*PKD1*	c.12386T>A	p.Met4129Lys	US	Unreported	Unreported	rs750629950	0.0000319	NA	1	APF (0.00)
Pt 6	*PKD2*	c.362delG	p.Gly121AlafsTer11	P	Unreported	Unreported	NA	NA	NA	NA	NA
Pt 9	*PKD1*	c.11459_11460insATGACAGCGGGGGCTACGT	p.Gln3821Ter	P	Unreported	Unreported	NA	NA	NA	NA	NA
Pt 10	*PKD1*	c.9559_9561delGAC	p.Asp3187del	LP	Unreported	LP	NA	NA	NA	NA	NA
Pt 10	*PKD1*	c.7825A>G	p.Ile2609Val	US	Unreported	Unreported	rs767709756	0.00002582	0.00107	0.566	T (0.15)
Pt 11	*PKD1*	c.6290G>T	p.Gly2097Val	US	Unreported	Unreported	NA	NA	NA	1	APF (0.00)
Pt 12	*PKD1*	c.6561G>A	p.Trp2187Ter	P	Unreported	Unreported	NA	NA	NA	NA	NA
Pt 15	*PKD1*	c.3161+5G>A	p.?	US	US	Unreported	NA	NA	NA	NA	NA
Pt 16	*PKD1*	c.10859_10860insA	p.Arg3621AlafsTer6	P	Unreported	Unreported	NA	NA	NA	NA	NA
Pt 17	*PKD1*	c.10822-3C>G	p.?	US	Unreported	Unreported	NA	NA	NA	NA	NA
Pt 18	*PKD1*	c.3161+5G>A	p.?	US	US	Unreported	NA	NA	NA	NA	NA
Pt 20	*PKD1*	c.12734G>C	p.Ser4245Thr	US	Unreported	Unreported	NA	NA	NA	0.47	APF (0.00)
Pt 21	*PKD1*	c.4796_4797insA	p.Tyr1599Ter	P	Unreported	Unreported	NA	NA	NA	NA	NA
Pt 22	*PKD1*	c.4306C>T	p.Arg1436Ter	P	P	P	rs1567200516	NA	NA	NA	NA
Pt 23	*PKD1*	c.5830G>T	p.Gly1944Ter	P	P	Unreported	rs200001471	0.000157	NA	NA	NA
Pt 24	*PKD1*	c.6643C>T	p.Arg2215Trp	LP	P1, LP2, US1	LP	rs752793757	0.000008359	NA	0.999	APF (0.02)
Pt 26	*GANAB*	c.203A>G	p.Gln68Arg	US	Unreported	Unreported	rs150062507	0.000251	0.00471	0.001	T (0.23)
Pt 27	*PKD2*	c.709+1_709+4delGTAA	p.?	P	Unreported	Unreported	NA	NA	NA	NA	NA
Pt 30	*PKD1*	c.6421G>T	p.Val2141Phe	US	Unreported	Unreported	NA	NA	NA	1	APF (0.01)
Pt 31	*PKD1*	c.6882_6883delCA	p.Ser2295PhefsTer124	P	Unreported	P	NA	NA	NA	NA	NA
Pt 32	*PKD1*	c.11441_11459dup	p.Gln3821Ter	P	Unreported	Unreported	NA	NA	NA	NA	NA
Pt 33	*PKD2*	c.2224C>T	p.Arg742Ter	P	P	P	rs121918040	0.00003185	NA	NA	NA
Pt 34	*PKD2*	c.1549-1G>C	p.?	P	Unreported	Unreported	rs1720121027	NA	NA	NA	NA
Pt 36	*PKD2*	c.1717-1G>C	p.?	P	Unreported	Unreported	NA	NA	NA	NA	NA
Pt 40	*PKD2*	c.2195_2205delGAGGAGGCAAG	p.Gly732ValfsTer4	P	Unreported	Unreported	NA	NA	NA	NA	NA
Pt 42	*PKD1*	c.4796_4797insA	p.Tyr1599Ter	P	Unreported	Unreported	NA	NA	NA	NA	NA
Pt 45	*PKD1*	c.7825A>G	p.Ile2609Val	US	Unreported	Unreported	rs767709756	0.00002582	0.00107	0.566	T (0.15)
Pt 47	*PKD1*	c.9806G>A	p.Arg3269Gln	US	Unreported	Unreported	rs550467690	0.000005862	NA	1	APF (0.03)

ACMG, American College of Medical Genetics and Genomics; APF, affect protein function; dbSNP, Single Nucleotide Polymorphism Database; del, deletion; gnomAD, Genome Aggregation Database; LOVD, Leiden Open Variation Database; LP, likely pathogenic; NA, not available; P, pathogenic; PolyPhen-2, Polymorphism Phenotyping v2; Pt, patient; SIFT, Sorting Intolerant From Tolerant; T, tolerated; US, uncertain significance.

**Table 3 genes-14-00443-t003:** Exome analyses of cyst-associated genes.

Pt	Gene	Variants	Amino Acid Change	ACMG	ClinVar	LOVD	dbSNP	gnomAD	3.5KJPNv2	PolyPhen-2 (Score)	SIFT (Score)
Pt 7	*BBS12*	c.775A>G	p.Thr259Ala	US	US	Unreported	rs746565072	0.00001989	0.0037	0	T (0.61)
Pt 7	*OFD1*	c.1215A>C	p.Glu405Asp	US	Unreported	Unreported	rs772129129	0.000005478	NA	0.675	NA
Pt 13	*IFT140*	c.2317C>T	p.Arg773Trp	US	Unreported	US	rs202236303	0.00002386	0.00018	0.999	NA
Pt 13	*NPHP3*	c.2986G>A	p.Val996Met	US	US	US	rs150867534	0.000131	0.00841	0.537	T (0.52)
Pt 19	*ZNF423*	c.955G>A	p.Ala319Thr	US	US	US	rs199919703	0.00009351	NA	0.981	T (0.19)
Pt 28	*PKHD1*	c.9629C>G	p.Ser3210Cys	US	Conflicting	Unreported	rs141081295	0.0001735	0.06462	1	T (0.11)
Pt 28	*TMEM67*	c.1243G>A	p.Val415Met	US	Unreported	Unreported	NA	NA	NA	0.986	NA
Pt 35	*GLIS2*	c.1403C>T	p.Thr468Met	US	Conflicting	Unreported	rs138285254	0.0001137	0.01749	0.999	APF (0.00)
Pt 35	*PKHD1*	c.1481G>A	p.Arg494Gln	US	Unreported	Unreported	rs151070471	0.00005661	NA	0.996	APF (0.05)
Pt 35	*WDR19*	c.3262-2A>G	p.?	P	LP	Unreported	rs753291151	0	0.00036	NA	NA
Pt 37	*WDPCP*	c.151C>T	p.His51Tyr	US	US	Unreported	rs779689937	0.00002011	0.0028	0.949	NA
Pt 38	*DNAJB11*	c.672delT	p.Phe224LeufsTer39	LP	Unreported	Unreported	NA	NA	NA	NA	NA
Pt 41	*NEK8*	c.1864delG	p.Asp622MetfsTer5	LP	Unreported	Unreported	NA	NA	NA	NA	NA
Pt 43	*NPHP3*	c.2986G>A	p.Val996Met	US	US	US	rs150867534	0.000131	0.00841	0.537	T (0.52)
Pt 43	*SLC41A1*	c.509A>G	p.Lys170Arg	US	Unreported	Unreported	NA	NA	NA	0.014	T (0.80)
Pt 46	*PKHD1*	c.865C>T	p.Gln289Ter	P	P	Unreported	NA	NA	NA	NA	NA
Pt 48	*PKHD1*	c.9629C>G	p.Ser3210Cys	US	Conflicting	Unreported	rs141081295	0.0001735	0.06462	1	T (0.11)
Pt 49	*KIF7*	c.2476C>T	p.Arg826Trp	US	US	Unreported	rs139711238	0.0001031	0.00084	1	APF (0.00)
Pt 50	*CEP164*	c.3557T>C	p.Leu1186Pro	US	Unreported	Unreported	NA	NA	NA	1	NA

ACMG, American College of Medical Genetics and Genomics; APF, affect protein function; dbSNP, Single Nucleotide Polymorphism Database; del, deletion; gnomAD, Genome Aggregation Database; LOVD, Leiden Open Variation Database; LP, likely pathogenic; NA, not available; P, pathogenic; PolyPhen-2, Polymorphism Phenotyping v2; Pt, patient; SIFT, Sorting Intolerant From Tolerant; T, tolerated; US, uncertain significance.

**Table 4 genes-14-00443-t004:** Comorbidity evaluation.

Pt	HT	BNP (pg/mL)	EF (%)	AR	MR	TR	PR	Septal e’ (cm/s)	Septal E/e’	TRV (m/s)	LAVI (mL/m^2^)	Liver Cysts	Brain MRA	PROPKD (Score)
Pt 1	(+)	5.8	62	(−)	(−)	Mi	(−)	6.9	7.8	2.2	NA	(−)	normal	5
Pt 2	(+)	6.1	69	T	T	T	Mi	5.7	9.6	2.0	60	(+)	rt. VA AN	0
Pt 3	(+)	440	51	(−)	T	(−)	(−)	5.1	12.4	NA	NA	(+)	normal	3
Pt 4	(+)	48.4	65	(−)	T	T	(−)	4.8	12.0	2.0	NA	(+)	normal	0
Pt 5	(+)	21	65	(−)	T	T	T	6.9	8.4	0.7	21.0	(+)	lt. ICA AN	2
Pt 6	(+)	120	70.1	(−)	(−)	(−)	Mi	7.7	7.9	NA	NA	(+)	normal	1
Pt 7	(+)	42.1	75.6	(−)	Mi	Mi	Mi	NA	NA	NA	NA	(−)	normal	1
Pt 8	(+)	27.7	70.8	Mo	Mi	Mi	Mi	6.6	8.7	NA	NA	(++)	normal	4
Pt 9	(+)	7.3	68	(−)	T	T	T	5.4	8.7	1.4	23	(+)	lt. MCA occlusion	4
Pt 10	(+)	15.2	67.5	(−)	(−)	Mi	Mi	6.6	8.4	NA	NA	(++)	normal	3
Pt 11	(+)	14.3	76.5	(−)	Mi	Mi	(−)	6.2	7.9	NA	NA	(+)	lt. ICA AN	2
Pt 12	(+)	6	69	(−)	T	T	T	4.3	7.6	NA	16.2	(++)	lt. MCA AN	6
Pt 13	(−)	23.8	69.7	(−)	(−)	Mi	T	8.2	7.4	NA	27.2	(+)	normal	1
Pt 14	(+)	27.6	70.1	(−)	Mi	Mi	Mi	7.7	9.5	NA	NA	(+)	rt. A1 perforator dilatation	1
Pt 15	(+)	52.1	72.2	(−)	Mi	Mi	(−)	12.1	5.3	NA	NA	(+)	lt. IC−PC dilatation	2
Pt 16	(+)	20.2	64.6	(−)	Mi	Mi	Mi	6.2	10.1	NA	NA	(+)	normal	4
Pt 17	(+)	84.1	62.1	(−)	Mi	Mi	(−)	8.6	7.5	NA	NA	(+)	normal	2
Pt 18	(+)	20	70.4	(−)	Mi	Mi	Mi	7.1	6.9	NA	NA	(+)	lt. ICA dilatation	2
Pt 19	(+)	105.6	63.8	Mi	Mi	Mi	Mi	5.2	19.3	NA	NA	(+)	lt. MCA AN	0
Pt 20	(+)	10.3	78.5	(−)	(−)	(−)	(−)	7.1	9.4	NA	NA	(+)	normal	3
Pt 21	(+)	26.5	75.6	(−)	(−)	Mi	Mi	9.1	4.8	NA	NA	(+)	normal	4
Pt 22	(+)	23.2	79.4	(−)	Mi	Mi	Mi	9.7	9.7	NA	NA	(++)	normal	4
Pt 23	(+)	2225.5	59.4	Mo	(−)	(−)	Mi	NA	NA	NA	NA	(+)	normal	5
Pt 24	(+)	20.7	66.7	Mo	(−)	(−)	Mi	7.5	9.0	NA	NA	(+)	normal	2
Pt 25	(+)	37	60.5	Mo	T	(−)	(−)	6.8	5.5	NA	NA	(+)	normal	1
Pt 26	(+)	133.6	73.1	Mi	Mi	Mi	Mi	5.2	9.3	NA	NA	(++)	normal	0
Pt 27	(+)	11.1	72.8	Mi	Mi	Mo	Mi	8.6	8.6	NA	NA	(+)	normal	1
Pt 28	(−)	6.3	74.9	(−)	(−)	(−)	(−)	8.1	6.2	NA	NA	(+)	lt. M1 stenosis	0
Pt 29	(+)	6.2	73.2	(−)	Mi	Mi	(−)	5.4	8.1	NA	NA	(+)	normal	5
Pt 30	(−)	39.3	76.8	(−)	(−)	Mi	(−)	14.7	6.5	2.1	NA	(−)	normal	2
Pt 31	(−)	33.9	69.9	(−)	Mi	Mi	Mi	9.9	8.9	2.0	NA	(+)	normal	4
Pt 32	(−)	15.3	62.4	(−)	(−)	(−)	(−)	7.8	6.6	NA	NA	(++)	normal	4
Pt 33	(−)	43	68.9	Mi	T	T	T	8.0	7.0	NA	27.2	(+)	normal	0
Pt 34	(−)	12.2	77	T	Mi	T	(−)	9.5	6.2	NA	19	(+)	Acom AN	0
Pt 35	(+)	113.6	65	Mi	Mi	T	T	3.5	15.5	NA	32	(−)	rt. MCA AN	0
Pt 36	(−)	35.4	73	T	Mi	Mi	(−)	7.9	9.9	2.4	45	(+)	normal	0
Pt 37	(+)	36.9	73	T	(−)	Mi	T	5.2	9.6	2.4	13	(−)	normal	1
Pt 38	(+)	NA	76.8	Mi	(−)	(−)	(−)	NA	NA	NA	NA	(+)	Acom perforator dilatation	0
Pt 39	(+)	32.9	65.1	(−)	(−)	Mi	(−)	8.5	8.5	2.2	NA	(+)	normal	0
Pt 40	(+)	11	70	T	T	Mi	(−)	6.4	9.5	2.2	24	(+)	normal	1
Pt 41	(+)	13.9	77.6	(−)	Mi	Mi	Mi	8.2	10.9	2.3	NA	(+)	normal	0
Pt 42	(+)	24.2	50	(−)	Mi	Mi	(−)	8.8	6.5	NA	NA	(+)	normal	5
Pt 43	(+)	5.4	62.2	(−)	Mi	Mi	(−)	6.3	13.1	NA	NA	(+)	normal	1
Pt 44	(−)	5.8	72.5	Mi	Mi	Mi	(−)	9.9	6.5	1.6	NA	(+)	normal	4
Pt 45	(−)	53	78.1	Mo	Mi	S	Mi	7.1	13.3	NA	NA	(−)	lt. PCA stenosis, rt. M1 stenosis	0
Pt 46	(+)	17	69	T	T	T	T	7.3	15.9	2.3	27	(−)	normal	1
Pt 47	(+)	NA	73.5	(−)	(−)	Mi	Mi	NA	NA	NA	NA	(+)	normal	3
Pt 48	(+)	24.4	71	(−)	Mi	Mi	Mi	6.7	6.9	1.9	NA	(++)	normal	0
Pt 49	(+)	32	60.5	(−)	(−)	T	(−)	4.8	10.8	NA	NA	(+)	normal	0
Pt 50	(+)	23.7	63	(−)	(−)	(−)	(−)	12.3	7.7	NA	NA	(+)	rt. MCA AN	0

Acom, anterior communicating artery; AN, aneurysm; AR, aortic valve regurgitation, BNP, brain natriuretic peptide; EF, ejection fraction; HT, hypertension; ICA, internal carotid artery; IC-PC, internal carotid-posterior communicating artery; LAVI, left atrial volume index; lt, left; MCA, middle cerebral artery; Mi, mild; Mo, moderate; MR, mitral valve regurgitation; MRA, magnetic resonance angiography; NA, not available; PCA, posterior cerebral artery; PR, pulmonic valve regurgitation; PROPKD, Predicting Renal Outcome in Polycystic Kidney Disease; rt, right; S, severe; T, trivial; TR, tricuspid valve regurgitation; TRV, tricuspid regurgitant velocity; VA, vertebral artery.

**Table 5 genes-14-00443-t005:** Changes in eGFR and TKV in patients managed without and with tolvaptan.

Pt	Tolvaptan	eGFR Change per Year (mL/min/1.73 m^2^)	Initial TKV	TKV Change per Year (%)
Pt 6	(−)	−4.3	2192	3.8
Pt 7	(−)	−0.1	810	1.9
Pt 12	(−)	−6.0	1331	25.6
Pt 13	(−)	−1.0	2130	1.7
Pt 19	(−)	−2.1	526	−0.3
Pt 26	(−)	−1.6	713	3.1
Pt 28	(−)	−1.4	601	−0.6
Pt 31	(−)	−3.9	396	−2.3
Pt 37	(−)	−0.7	1576	−2.2
Pt 38	(−)	−2.5	415	2.2
Pt 39	(−)	−2.2	293	2.5
Pt 41	(−)	−3.6	811	2.8
Pt 43	(−)	−2.8	1695	12.7
Pt 45	(−)	−4.4	867	8.5
Pt 47	(−)	−2.6	1004	3.4
Pt 48	(−)	−2.0	1580	−1.5
Pt 1	(+)	−9.9	986	9.1
Pt 5	(+)	2.4	1064	−4.9
Pt 8	(+)	−2.0	801	3.8
Pt 9	(+)	−2.4	2023	9.9
Pt 10	(+)	−2.4	2690	7.4
Pt 11	(+)	−1.9	1320	2.0
Pt 14	(+)	−3.1	1078	17.6
Pt 15	(+)	−4.7	1687	13.0
Pt 16	(+)	−2.8	1837	3.0
Pt 17	(+)	−2.7	1010	−2.6
Pt 18	(+)	−4.6	1046	5.6
Pt 20	(+)	−2.0	1611	2.6
Pt 21	(+)	−1.6	1344	1.0
Pt 22	(+)	−3.5	1429	3.6
Pt 23	(+)	−3.1	2487	9.1
Pt 24	(+)	−4.3	3219	7.4
Pt 27	(+)	−0.4	1351	3.5
Pt 29	(+)	−5.0	1340	9.5
Pt 32	(+)	−3.4	848	4.2
Pt 35	(+)	−1.6	1412	6.0
Pt 42	(+)	−9.3	1110	11.8
Pt 44	(+)	0.2	700	3.2
Pt 46	(+)	−3.9	891	−2.5
Pt 50	(+)	−2.4	1021	−4.0

eGFR, estimated glomerular filtration rate; Pt, patient; TKV, total kidney volume.

## Data Availability

The data presented in this study are available on request from the corresponding author. The data are not publicly available due to ethical restrictions.

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
