# Peer review of "Mutation Analysis of Autosomal-Dominant Polycystic Kidney Disease Patients"

_genes, 2023, doi:10.3390/genes14020443_

Round 1

Reviewer 1 Report

Suzuki et al investigated the genetic variants underlying autosomal dominant polycystic kidney disease with 50 samples by using exome sequencing or multiplex ligation-dependent probe amplification, followed by long polymerase chain reaction and Sanger sequencing. They identified many genetic variants at PKD1 and PKD2, two main causative genes of ADPKD, some of which are novel variants. Besides, they identified several genetic variants at other genes. The study expanded the variant spectrum underlying ADPKD. The sample size are large.

I have one major concern. Since you applied the exome sequencing technology, you were able to analyze the genetic variants at almost every gene on human genome. But in the analysis, you only focused on 93 cyst-associated genes as you described in the method section (lines 67-76). Please clarify. If exome data were used, genetic variants at other genes on human genome should also be provided. This is important as your previous strategy may miss true causative variants or novel genes underlying ADPKD.

Author Response

Response

Thank you for your suggestion. As suggested, we searched all genes by exome sequencing and have provided rare pathogenic variants of other genes in 15 patients (patients 7, 13, 19, 25, 28, 35, 37, 38, 39, 41, 43, 46, 48, 49, 50) in Table S6. There were no rare pathogenic variants of other genes in patients 28, 38 or 39. In the same way, we identified rare pathogenic variants of other genes in 11 patients (patients 1, 5, 11, 15, 17, 18, 20, 26, 30, 45, 47) with variants of uncertain significance in PKD1 or PKD2 or GANAB shown in Table S7. There were no rare pathogenic variants in other genes in patient 26. We have added this information to the Results section at page 10, line 182-188.

Reviewer 2 Report

1.   In Table  1,  PROPKD score calculation needs PKD1/PKD2 genetic information.  How do you calculate the PROPKD score in undiagnosed/US or non-PKD1/PKD2 individuals? Putting the PROPKD score in Table 1 may not be appropriate.

2. In Table 1, the family history column contains numbers, an explanation should be described  (number of other affected members?)

3. The manuscript should mention that pathogenic variants were identified in 24 patients (15 from  PKD1/PKD2 ES, 5 from MLPA, and 4 from 90 genes ES)  in the current cohort. Others have variants of uncertain significance, which do not indicate a definite diagnosis until further proof.

4. Two variants are found in patient 1 and patient 10. Are those variants in-cis or in-trans? This can be done if their parents or siblings are available.  

5. Patients with variants of US should not be considered as a final correct genetic diagnosis.   MLPA and other 90 genes should be analyzed in US patients.

6. All ALG genes, IFT-related genes,  and COL4A genes should be included in the exome sequencing analysis since a significant portion of patients has atypical cysts (Mayo classification type II) and proteinuria in the cohort.

7. A significant number of patients (17 patients) are diagnosed with Mayo classifications type II. 9  have pathogenic variants, including all 5 PKD1/PKD2  deletions and 4 pathogenic variants (DNAJB11, PKHD1, NEK8, and WDR19).  Their kidney images should be provided for the interest and compare if possible.

8.  DNAJB11 is a known disease-causing gene of ADPKD (https://www.ncbi.nlm.nih.gov/pmc/articles/PMC5986722/) and kidney cysts can develop in patients of heterozygous PKHD1 pathogenic variant. Whether heterozygous NEK8 and WDR19 pathogenic variants can mimic PKD phenotype need more evidence. 

Author Response

  1. In Table 1, PROPKD score calculation needs PKD1/PKD2 genetic information. How do you calculate the PROPKD score in undiagnosed/US or non-PKD1/PKD2 individuals? Putting the PROPKD score in Table 1 may not be appropriate.

Response

As suggested, we have described the PROPKD score in Table 4 and Table S8.

  1. In Table 1, the family history column contains numbers, an explanation should be described (number of other affected members?)

Response

As suggested, we have added “and the number of other affected members is shown in Table 1” at page 3, line 115.

  1. The manuscript should mention that pathogenic variants were identified in 24 patients (15 from PKD1/PKD2 ES, 5 from MLPA, and 4 from 90 genes ES) in the current cohort. Others have variants of uncertain significance, which do not indicate a definite diagnosis until further proof.

Response

As suggested, we have added the sentence "Overall, “likely pathogenic” or “pathogenic” variants were identified in 28 patients; these were identified by PKD1/PKD2 exome analyses in 19 patients; MLPA analyses in 5 patients; and exome sequencing of 90 genes in 4 patients. Other patients had variants of uncertain significance, which did not indicate a definite diagnosis until further proof was obtained" in the Discussion section at page 17, line 261-264.

  1. Two variants are found in patient 1 and patient 10. Are those variants in-cis or in-trans? This can be done if their parents or siblings are available.

Response

Since we could not obtain blood samples from parents or siblings of patient 1 or 10, we could not examine whether those variants were in-cis or in-trans. We have added “While two variants in PKD1 were found in patients 1 and 10, respectively, we could not examine whether these acted as in cis or in trans variants, as blood samples could not be obtained from parents or siblings” as a limitation in the Discussion section at page 18, line 314-317.

  1. Patients with variants of US should not be considered as a final correct genetic diagnosis. MLPA and other 90 genes should be analyzed in US patients.

Response

Variants in patients 2, 3, and 24 were considered to be “likely pathogenic” after reevaluation according to the ACMG guidelines. As suggested, we have performed an MLPA analysis in 10 US patients (patients 5, 11, 15, 17, 18, 20, 26, 30, 45, 47). An MLPA analysis was not performed for patient 1 because patient 1 had two variants in PKD1 (one variant was an in-frame deletion). The MLPA analyses showed negative results in the 10 US patients. We searched all genes by exome sequencing and identified rare pathogenic variants in other genes in 11 patients (patients 1, 5, 11, 15, 17, 18, 20, 26, 30, 45, 47); these are listed in Table S7. There were no rare pathogenic variants in other genes in patient 26. We have added this information to the Results section at page 10, line 185-188.

  1. All ALG genes, IFT-related genes, and COL4A genes should be included in the exome sequencing analysis since a significant portion of patients has atypical cysts (Mayo classification type II) and proteinuria in the cohort.

Response

As suggested, we searched all genes by exome sequencing and identified rare pathogenic variants in other genes, including all ALG genes, IFT-related genes, and COL4A genes. The results are ere shown in Tables S6 and S7.

  1. A significant number of patients (17 patients) are diagnosed with Mayo classifications type II. 9 have pathogenic variants, including all 5 PKD1/PKD2 deletions and 4 pathogenic variants (DNAJB11, PKHD1, NEK8, and WDR19). Their kidney images should be provided for the interest and compare if possible.

Response

As suggested, 18 patients (patients 7, 8, 13, 14, 19, 28, 29, 35, 37-39, 41, 43, 44, 46, 48-50) had a type II Mayo classification. Eight patients (patients 8, 14, 29, 35, 38, 41, 44, 46) had pathogenic variants, including four PKD1/PKD2 deletions in patients 8, 14, 29, 44, and four pathogenic variants (DNAJB11, NEK8, PKHD1, and WDR19). We have provided kidney images of the 8 pathogenic variants in Figure 2. We have added this information to the Results section at page 12, line 218-225.

  1. DNAJB11 is a known disease-causing gene of ADPKD (https://www.ncbi.nlm.nih.gov/pmc/articles/PMC5986722/) and kidney cysts can develop in patients of heterozygous PKHD1 pathogenic variant. Whether heterozygous NEK8 and WDR19 pathogenic variants can mimic PKD phenotype need more evidence.

Response

As suggested, we have added “While DNAJB11 is a known disease-causing gene of ADPKD [25] and kidney cysts can develop in patients who are heterozygous for a pathogenic variant of PKHD1, more evidence is required to support whether heterozygous NEK8 and WDR19 pathogenic variants can mimic the ADPKD phenotype.” as a limitation in the Discussion section at page 18, line 317-321.

Round 2

Reviewer 1 Report

Yasuo Suzuki et al addressed my concern. Why other gene harboring functional variants were not discussed in the main text? Just providing a supplementary table is not enough to highlight the novelty of your findings.

Author Response

To Reviewer 1

Yasuo Suzuki et al addressed my concern. Why other gene harboring functional variants were not discussed in the main text? Just providing a supplementary table is not enough to highlight the novelty of your findings.

Response: As suggested, we have added, “While rare pathogenic variants in other genes were examined in 15 patients who showed negative results in exome sequencing or MLPA analyses, their functional roles in ADPKD were undetermined. Rare pathogenic variants in other genes were also examined in 11 patients with variants of uncertain significance in PKD1, PKD2, or GANAB in a similar way, and while the functional roles in ADPKD were undetermined for most of these variants, the roles of CPLANE1 in patient 18 and AHI1 in patient 47, both of which were cilia-associated genes, might be involved in development of ADPKD to some degree.” to the Discussion section at page 17, line 271-278. We also added the limitation, “Rare pathogenic variants in other genes were not validated in Sanger sequencing.” to the Discussion section at page 18, line 328-329.
